# Unveiling fatal risk factors: Predicting hemophagocytic lymphohistiocytosis in SFTS patients

**Bo Zhang**(iD)*, **Jia-le Gong, Hao-long Zeng, Qin Liao**

Department of Laboratory Medicine, Tongji Hospital, Tongji Medical College, Huazhong University of Science and Technology, Wuhan, China

* zb280305@163.com

## Abstract

### Background

Severe fever with thrombocytopenia syndrome (SFTS) is a zoonotic infectious disease with high mortality, and hemophagocytic lymphohistiocytosis (HLH) is one of the rare fatal complications of SFTS. Early prediction of the occurrence of HLH and the identification of prognostic factors in SFTS patients with HLH are crucial for effective clinical management.

### Methods

Univariate and multivariate logistic regression were used to analyze the demographic characteristics, clinical manifestations at admission and laboratory parameters of 272 SFTS patients. The ROC curve was used to calculate the optimal critical value of each index based on the survival outcomes of patients, and the kinetic characteristics of laboratory markers predicting the prognosis of patients with HLH were analyzed.

### Results

Decreased platelet count, reduced ALT/AST ratio, elevated LDH, and increased DD were identified as independent risk factors for HLH in SFTS patients. Age, fibrinogen (FIB), and procalcitonin (PCT) were independent risk factors for mortality in SFTS patients with HLH ($P < 0.05$). The combination of these three factors can effectively predict patient prognosis (AUC = 0.903). Patients aged ≥64 years, with FIB ≤ 2.23 g/L, and PCT ≥ 0.9 ng/ml exhibited higher mortality rates. The dynamic characteristics of PCT and FIB levels significantly differed between the survival and death groups in SFTS patients with HLH.

**Data availability statement:** The data supporting the findings of this study are available within the manuscript and its Supporting Information files.

**Funding:** The author(s) received no specific funding for this work.

**Competing interests:** The authors have declared that no competing interests exist.

## Conclusion

Early laboratory indicators can timely identify HLH complications in SFTS patients. Close monitoring of elderly patients and regular assessment of PCT and FIB levels can effectively reduce mortality.

### Author summary

SFTS is a zoonotic disease with high mortality. While research has implicated HLH as a leading fatal complication of SFTS, the majority of these studies have focused on isolated cases, lacking comprehensive correlation analysis. This study identifies a decreased alanine aminotransferase/aspartate aminotransferase (ALT/AST) ratio, along with elevated levels of LDH and DD, as independent predictors of HLH development in patients with SFTS. Additionally, age ≥ 64 years, fibrinogen levels ≤2.23 g/L, and procalcitonin levels ≥0.9 ng/mL collectively serve as significant prognostic markers for mortality in cases of SFTS complicated by HLH. Continuous monitoring of fibrinogen and procalcitonin levels enhances the real-time assessment of disease progression, facilitating timely interventions to reduce mortality.

## Introduction

Severe Fever with Thrombocytopenia Syndrome (SFTS) is a zoonotic infectious disease caused by SFTS virus (SFTSV) [1–2]. The most common clinical symptoms of SFTS include fever, thrombocytopenia, leukopenia, and gastrointestinal abnormalities [3]. Since its initial report in China in 2010, the disease has been documented in various countries and regions worldwide [4]. The WHO has listed SFTSV as one of the nine most dangerous pathogens that require prompt attention [5]. In severe cases, SFTSV infection can lead to immune dysfunction, cytokine storm, and endothelial injury, potentially resulting in hemorrhage or multiple organ failure, with a mortality rate ranging from 2.8% to 47% [3,6]. Hemophagocytic lymphohistiocytosis (HLH) is a severe hyperinflammatory response syndrome triggered by abnormally activated macrophages and cytotoxic T cells. The disease progresses rapidly and can cause systemic tissue and organ damage within a short period. Without timely and effective intervention, the prognosis for patients is extremely poor [7]. Previous studies have shown that HLH is one of the fatal complications of SFTS, and it is considered a key factor that increase the risk of death in SFTS patients [8–11]. However, these studies are mainly isolated case reports without any correlation analysis.Therefore, this study analyzed the demographic characteristics, clinical features and laboratory parameters of SFTS patients with HLH at admission, and examined the early predictive factors and prognostic factors of patients with HLH, to promote the early detection of HLH complications and improve the survival rate of patients.

## Materials and methods

### Ethics statement

This study was approved by the Ethics Committee of Tongji Hospital of Huazhong University of Science and Technology (NO.TJ-IRB20230632). The study was performed in accordance with the Declaration of Helsinki. Due to the retrospective design, patient informed consent was waived.

### Patients

We gathered hospitalized SFTS patients at Tongji Hospital, Tongji Medical College, Huazhong University of Science and Technology from January 2020 to April 2024. SFTSV infection was confirmed by detecting SFTSV RNA in EDTA-anticoagulated plasma through polymerase chain reaction or next-generation sequencing. Patients with co-infections of hepatitis C virus, hepatitis B virus, or human immunodeficiency virus (HIV), as well as those with malignant tumors or autoimmune diseases, were excluded.

### Collection and analysis of clinical data

Clinical data were collected through the Medical Data Retrieval and Application Platform of Tongji Hospital in Wuhan, China. The data included demographic information, underlying diseases, clinical symptoms, laboratory test results, treatment methods, and prognosis. Laboratory tests included routine blood markers, biochemical indicators, inflammatory markers, cardiac markers, coagulation markers, and cytokines. Routine blood markers were determined using the Sysmex XN 2000 automated hematology analyzer.

The analysis of blood biochemical markers was conducted utilizing the Roche fully automated biochemical analyzer, encompassing liver function indicators—alanine aminotransferase (ALT), aspartate aminotransferase (AST), total protein (TP), albumin (ALB), total bilirubin (TBIL), direct bilirubin (DBIL), alkaline phosphatase (ALP), gamma-glutamyl transferase (GGT), total cholesterol (T-CHOL), and lactate dehydrogenase (LDH)—as well as kidney function indicators, including urea, creatinine, uric acid, and bicarbonate. Furthermore, specific inflammatory markers, namely interleukin-6 (IL-6), high-sensitivity C-reactive protein (hsCRP), procalcitonin (PCT), and ferritin, were also quantified. Some cardiac markers were detected using the Abbott chemical analyzer. Coagulation parameters including prothrombin time (PT), activated partial thromboplastin time (APTT), thrombin time (TT), fibrinogen (FIB), and D-dimer (DD) were measured using Stago coagulation analyzers. Serum levels of interleukin-1β (IL-1β), interleukin-2 receptor (IL-2R), interleukin-8 (IL-8), interleukin-10 (IL-10), and tumor necrosis factor-alpha (TNF-α) were determined using the SIEMENS Immulite 1000 solid-phase two-site chemiluminescent immunoassay. All laboratory test results from the time of admission to discharge were collected for all confirmed SFTS patients.

### Diagnosis of HLH

The diagnosis of HLH was based on HLH-2004 criteria [12], requiring the presence of at least 5 out of the following 8 criteria: (1) Fever: body temperature > 38.5°C, lasting > 7 days; (2) Splenomegaly; (3) Cytopenia (involving two or three lineages of peripheral blood): hemoglobin < 90 g/L, platelets < 100 × 10^9/L, neutrophils < 1.0 × 10^9/L, and not due to decreased bone marrow hematopoiesis; (4) Hypertriglyceridemia and/or hypofibrinogenemia: triglycerides >3 mmol/L or above the age-matched 3 standard deviations, fibrinogen <1.5 g/L or below the age-matched 3 standard deviations; (5) Hemophagocytes found in bone marrow, spleen, liver, or lymph nodes; (6) Elevated serum ferritin: ferritin ≥ 500 μg/L; (7) Decreased or absent NK cell activity; (8) Elevated sCD25 (soluble interleukin-2 receptor).

### Statistical analysis

Statistical analyses were conducted using GraphPad Prism version 9.5 (San Diego, CA, USA) and SPSS version 26.0 (Chicago, IL, USA). Normally distributed continuous data were presented as mean ± standard deviation (SD)

and compared between two groups using Student's t-test. Non-normally distributed continuous data were shown as median (IQR) and compared using the Mann-Whitney test. Categorical data were expressed as frequency (%) and compared using the Chi-square test. Univariate analyses assessed differences in demographic characteristics, clinical manifestations, and laboratory parameters for HLH occurrence. Factors with $P < 0.05$ in univariate analysis were further evaluated as independent risk factors for HLH using multivariable logistic regression. Kaplan-Meier curves and log-rank tests were employed for survival analysis. Receiver operating characteristic (ROC) curves determined maximum values, sensitivity, specificity, and optimal cutoff values. Univariate and multivariate logistic regression models analyzed prognostic factors in SFTS patients with concurrent HLH. Statistical significance was set at $P < 0.05$.

## Results

### Demographic and clinical characteristics of SFTS patients

A total of 272 SFTS patients were included in this study, with 148 females and 124 males. The median age was 65 (IQR,57-71) years. Among these patients, 53 were in the HLH group and 219 were in the non-HLH group. The demographic and clinical characteristics of the SFTS patients are shown in **Table 1**. There were no significant differences in age and gender between the two groups. Compared to non-HLH SFTS patients, those in the HLH group had fewer diarrhea symptoms upon admission ($P = 0.037$). However, the onset time for patients in the HLH group was shorter ($P = 0.011$), and they required more treatments such as corticosteroids ($P = 0.013$), intravenous immunoglobulin ($P = 0.01$), and continuous renal replacement therapy ($P < 0.001$). Additionally, the mortality rate was higher in the HLH group ($P = 0.026$).

### Laboratory parameter analysis of SFTS patients at admission

The laboratory parameter results of SFTS patients at admission are presented in **Table 2**. Compared to non-HLH patients, SFTS patients with HLH exhibited significant differences in hematological markers, liver function markers, coagulation markers, inflammatory indicators, and cytokine markers. Hematological parameters showed that SFTS patients in the HLH group had lower counts of lymphocytes, monocytes, white blood cells, and platelets than those in the non-HLH group ($P < 0.05$). Biochemical markers indicated that SFTS patients with HLH had higher levels of ALT, AST, and LDH, but a lower ALT/AST ratio compared to those without HLH. Coagulation indicators revealed significant differences in APTT, TT, FIB, and DD between the two groups ($P < 0.05$), with FIB being lower in the HLH group. Inflammatory markers such as ferritin and PCT levels were significantly higher in the HLH group compared to the non-HLH group ($P < 0.001$), and cytokines IL6, IL2R, and IL10 were also elevated in the HLH group. These findings suggest that the differences in these markers may be associated with secondary HLH in SFTS patients.

### Independent risk factor analysis for HLH in SFTS patients

To identify independent risk factors for HLH occurrence in SFTS patients, we conducted a univariate logistic regression analysis of demographic characteristics, clinical presentations at admission, and laboratory parameters between the two groups. The analysis revealed that the time from onset to admission, diarrhea, platelet count, ALT/AST ratio, LDH, APTT, FIB, DD, ferritin, and IL-10 were significant factors associated with HLH in SFTS patients. Further multivariate logistic regression analysis indicated that a reduced platelet count (OR 0.966 [0.947, 0.985], $P = 0.001$), decreased ALT/AST ratio (OR 0.016 [0.001, 0.338], $P = 0.008$), elevated LDH (OR 1.001 [1.001, 1.002], $P = 0.001$), and increased DD (OR 1.088 [1.033, 1.146], $P = 0.001$) are independent risk factors for HLH occurrence in SFTS patients. Diarrhea was identified as a protective factor against HLH in SFTS patients (OR 0.434 [0.211, 0.895], $P = 0.024$). Although ferritin levels differed between the two groups ($P=0.002$), there was no significant correlation between ferritin levels and HLH occurrence in SFTS patients (OR 1.000). Specific results are shown in **Table 3**.

PLOS Neglected Tropical Diseases

**Table 1. Demographic and clinical features of SFTS patients on admission.**

| Parameters | Total (*n* = 272) | Non-HLH group (*n* = 219) | HLH group (*n* = 53) | *P*-value |
|---|---|---|---|---|
| **Sex, N (%)** | | | | 0.797 |
| Female | 148(54.4) | 120(54.8) | 28(52.8) | |
| Male | 124(45.6) | 99(45.2) | 25(47.2) | |
| **Age, Y, median (IQR)** | 65.0 (57.0,71.0) | 66.0 (57.0,70.0) | 64.0 (56.0,71.0) | 0.598 |
| **Area, N (%)** | | | | 0.400 |
| Urban area | 38 (14.0) | 33 (15.1) | 5 (9.4) | |
| Rural area | 234 (86.0) | 186 (84.9) | 48 (90.6) | |
| **Season of onset, N (%)** | | | | 0.875 |
| Spring and summer | 154 (56.6) | 125 (57.1) | 29 (54.7) | |
| Autumn and winter | 118 (43.4) | 94 (42.9) | 24 (45.3) | |
| **Symptoms, N (%)** | | | | |
| Fever | 242(89.0) | 192(87.7) | 50(94.3) | 0.164 |
| Muscular soreness | 56(20.6) | 47(21.5) | 9(17.0) | 0.469 |
| cough | 44(16.2) | 38(17.4) | 6(11.3) | 0.285 |
| Weakness | 129(47.4) | 108(49.3) | 21(39.6) | 0.205 |
| Inappetence | 87(32.0) | 68(31.1) | 19(35.9) | 0.502 |
| Nausea | 54(19.9) | 42(19.2) | 12(22.6) | 0.571 |
| Vomiting | 66(24.3) | 53(24.2) | 13(24.5) | 0.960 |
| Abdominal pain | 35(12.9) | 32(14.6) | 3(5.7) | 0.081 |
| Diarrhea | 106(39.0) | 92(42.0) | 14(26.4) | **0.037** |
| Headache and dizziness | 91(33.5) | 70(32.0) | 21(39.6) | 0.289 |
| Consciousness disorder | 15(5.5) | 12(5.5) | 3(5.7) | 0.959 |
| **Comorbidities, N (%)** | | | | |
| hypertension | 96(35.3) | 78(35.6) | 18(34.0) | 0.821 |
| Diabetes mellitus | 73(26.8) | 54(24.7) | 19(35.9) | 0.099 |
| Cerebrovascular diseases | 39(14.3) | 30(13.7) | 9(17.0) | 0.541 |
| Lung disease | 12(4.4) | 9(4.1) | 3(5.7) | 0.622 |
| Chronic kidney disease | 8(2.9) | 7(3.2) | 1(1.9) | 0.613 |
| **Therapy, N (%)** | | | | |
| Corticosteroid | 193(71.0) | 148(67.6) | 45(85.0) | **0.013** |
| Intravenous immunoglobulin | 88(32.3) | 63(28.8) | 25(47.2) | **0.010** |
| continuous renal replacement therapy | 75(27.6) | 49(22.4) | 26(49.1) | **<0.001** |
| Respiratory support | 48(17.7) | 37(16.9) | 11(20.8) | 0.508 |
| **History of tick bite, N (%)** | 81(29.8) | 68(31.1) | 13(24.5) | 0.352 |
| **Time from onset to admission,d,median (IQR)** | 7.0 (5.0,8.0) | 7.0 (5.0,9.0) | 6.0 (4.0,7.0) | **0.011** |
| **Duration of hospital admission,d,median (IQR)** | 8.0(4.0,12.0) | 7.0(5.0,12.0) | 9.0(4.0,13.0) | 0.497 |
| **Outcome, N (%)** | | | | 0.026 |
| Survival | 179(65.8) | 151(69.0) | 28(52.8) | |
| Deceased | 93(34.2) | 68(31.0) | 25(47.2) | |

## Prognostic analysis of SFTS patients with HLH

An analysis of the clinical characteristics of 272 SFTS patients indicated that the presence of HLH was statistically significantly associated with patient mortality. Kaplan-Meier curve analysis showed that the 28-day survival rate of SFTS

**Table 2. The comparison of laboratory parameters in SFTS patients with and without HLH.**

| Parameters | Total(n = 272) | Non-HLH group (n = 219) | HLH group (n = 53) | p |
|---|---|---|---|---|
| **Blood routine indicators** | | | | |
| WBC, × 10⁹/L | 3.16[1.81,5.93] | 3.29[1.89,6.01] | 2.45[1.61,4.21] | **0.044** |
| Lymphocyte, × 10⁹/L | 0.57[0.37,0.93] | 0.62[0.39,0.96] | 0.45[0.32,0.85] | **0.038** |
| Monocyte, × 10⁹/L | 0.13[0.08,0.33] | 0.15[0.08,0.37] | 0.09[0.06,0.13] | **0.002** |
| Neutrophil, × 10⁹/L | 1.94[1.15,4.26] | 2.06[1.18,4.65] | 1.72[1.00,3.33] | 0.168 |
| Platelet, × 10⁹/L | 47.0[32.0,64.0] | 50.0[33.0,68.0] | 33.0[25.0,52.0] | **<0.001** |
| RBC, × 10¹²/L | 4.29±0.68 | 4.31±0.69 | 4.22±0.67 | 0.400 |
| Hemoglobin,g/L | 129.36±20.55 | 129.64±20.40 | 128.23±21.09 | 0.655 |
| **Blood biochemistry indicators** | | | | |
| ALT, U/L | 86.0[52.0,162.0] | 80.0[48.0,151.0] | 116.0[70.0,193.0] | **0.011** |
| AST, U/L | 243.0[108.0,441.0] | 208.0[97.0,383.0] | 357.0[224.0,677.0] | **<0.001** |
| ALT/AST ratio | 0.38[0.28,0.54] | 0.39[0.29,0.56] | 0.32[0.22,0.45] | **0.016** |
| Total protein, g/L | 60.77±6.17 | 61.02±6.29 | 59.73±5.55 | 0.172 |
| Albumin, g/L | 32.56±4.76 | 32.73±4.82 | 31.87±4.43 | 0.239 |
| Albumin/globulin ratio | 1.18±0.24 | 1.18±0.24 | 1.17±0.25 | 0.843 |
| TBIL, μmol/L | 8.3[6.0,11.6] | 8.3[6.0,11.9] | 7.9[5.3,10.3] | 0.185 |
| DBIL, μmol/L | 4.3[3.0,6.7] | 4.4[3.1,6.8] | 3.8[2.9,6.5] | 0.256 |
| ALP, U/L | 74.0[58.0,108.0] | 73.0[58.0,109.0] | 75.0[58.0,98.0] | 0.910 |
| γ-Glutamyl transferase,U/L | 40.0[25.0,88.0] | 41.0[25.0,97.0] | 38.0[22.0,82.0] | 0.307 |
| LDH, U/L | 744.0[445.0,1282.0] | 727.0[425.0,1220.0] | 926.0[634.0,1474.0] | **0.008** |
| Potassium,mmol/L | 4.07[3.63,4.44] | 4.04[3.57,4.49] | 4.10[3.71,4.38] | 0.574 |
| Sodium,mmol/L | 134.1[130.8,136.6] | 134.4[130.8,136.9] | 133.0[130.4,134.4] | **0.011** |
| Chlorine,mmol/L | 100.7[97.6,103.0] | 100.8[97.8,103.2] | 100.29[96.4,102.5] | 0.157 |
| Calcium,mmol/L | 1.96[1.88,2.05] | 1.97[1.88,2.07] | 1.94[1.87,2.04] | 0.388 |
| Urea, mmol/L | 5.97[4.20,8.85] | 5.90[3.90,8.80] | 6.00[4.66,9.00] | 0.400 |
| Creatinine, μmol/L | 83.0[65.0,113.0] | 81.0[64.0,113.0] | 89.0[68.0,117.0] | 0.201 |
| Uric acid, mmol/L | 252.0[195.0,331.0] | 249.0[192.0,327.0] | 266.0[210.0,333.0] | 0.552 |
| HCO3⁻, mmol/L | 19.55±3.56 | 19.69±3.56 | 18.99±3.50 | 0.196 |
| eGFR, mL/min/1.73m² | 74.8[51.9,90.7] | 75.1[52.4,92.5] | 73.3[49.5,83.0] | 0.155 |
| Lactic acid,mmol/L | 1.62[1.27,2.18] | 1.59[1.26,2.17] | 1.74[1.27,2.20] | 0.532 |
| Glucose, mmol/L | 7.80[6.51,9.95] | 7.98[6.57,10.10] | 7.40[6.30,9.67] | 0.364 |
| Lipase, IU/L | 185.5[105.5,345.8] | 170.4[98.7,336.9] | 217.5[125.3,377.4] | 0.077 |
| Amylopsin, U/L | 88.0[54.0,132.0] | 85.0[52.3,131.0] | 96.0[62.0,147.0] | 0.320 |
| CK,U/L | 509.0[229.0,1371.0] | 509.0[206.0,1410.0] | 500.0[246.0,1371.0] | 0.731 |
| Triglyceride, mmol/L | 2.30[1.62,3.22] | 2.23[1.67,3.13] | 3.02[1.46,3.79] | 0.190 |
| Total cholesterol, mmol/L | 2.93[2.47,3.50] | 2.95[2.48,3.51] | 2.83[2.29,3.41] | 0.327 |
| **Coagulation markers** | | | | |
| PT,s | 12.9[12.3,13.7] | 12.9[12.2,13.7] | 13.2[12.6,14.2] | 0.077 |
| APTT,s | 56.6[46.3,68.4] | 55.1[45.1,66.6] | 62.4[54.7,85.7] | **<0.001** |
| TT,s | 24.6[20.8,35.1] | 23.9[20.7,31.7] | 31.9[22.2,50.6] | **0.003** |
| Fibrinogen, g/L | 2.52[2.23,3.03] | 2.58[2.29,3.05] | 2.36[2.10,2.75] | **0.024** |
| D-dimer, μg/mL FEU | 3.53[1.69,7.68] | 2.85[1.53,6.81] | 6.08[3.74,12.71] | **<0.001** |
| **Cardiac Markers** | | | | |
| Myoglobin,ng/mL | 207.0[112.5,508.7] | 207.0[115.6,505.3] | 189.5[109.2,567.6] | 0.974 |
| CK-MB,ng/mL | 3.3[1.7,7.5] | 3.3[1.8,7.7] | 3.1[1.3,7.5] | 0.607 |
| hs-cTnI,pg/mL | 107.3[32.8,295.1] | 111.2[33.2,347.8] | 72.6[29.4,211.3] | 0.144 |

*(Continued)*

| Parameters | Total(*n* = 272) | Non-HLH group (*n* = 219) | HLH group (*n* = 53) | *p* |
|---|---|---|---|---|
| NT-proBNP,pg/mL | 616.0[249.0,1405.0] | 640.4[259.9,1440.5] | 514.4[199.7,1244.0] | 0.360 |
| **Inflammatory indicators** | | | | |
| Ferritin, µg/L | 9986.4[3358.5,25591.0] | 8354.4[2705.7,22989.0] | 19733.0[7569.7,45124.0] | **<0.001** |
| hsCRP, mg/L | 6.3[2.0,16.6] | 5.9[1.8,16.8] | 7.7[4.2,13.1] | 0.138 |
| PCT, ng/mL | 0.40[0.14,1.15] | 0.33[0.13,0.93] | 0.90[0.42,2.00] | **<0.001** |
| **Cytokine indicators** | | | | |
| IL-6, pg/mL | 53.92[19.68,125.70] | 45.70[15.72,122.10] | 84.45[49.90,143.30] | **0.009** |
| IL-2R, U/mL | 1281.0[964.0,1856.0] | 1257.0[943.6,1739.0] | 1479.0[1138.0,2077.0] | **0.040** |
| IL-8, pg/mL | 43.4[21.4,117.0] | 38.180[20.5,114.0] | 51.0[25.6,128.0] | 0.403 |
| IL-10, pg/mL | 60.0[15.4,141.8] | 48.0[13.2,129.0] | 100.0[53.5,181.0] | **0.002** |
| IL-1β, pg/mL | 7.6[5.0,15.2] | 7.6[5.0,14.6] | 7.2[5.0,17.9] | 0.827 |
| TNF-α, pg/mL | 29.7[17.8,55.9] | 28.5[16.7,52.8] | 31.5[21.4,67.4] | 0.190 |

Abbreviations: ALP, alkaline phosphatase; ALT, alanine minotransferase; CK,Creatine phosphokinase;APTT, activated partial thromboplastin time; AST, aspartate aminotransferase; DBIL, direct bilirubin; eGFR, estimated glomerular filtration rate; FEU, fibrinogen equivalent unit; hsCRP, high-sensitivity C-reactive protein; IBIL, indirect bilirubin; IL, interleukin; LDH, lactate dehydrogenase; PCT, procalcitonin; PT, prothrombin time; RBC, red blood cell; TBIL, total bilirubin; TNF-α, tumor necrosis factor-α; TT, thrombin time; WBC, white blood cell;hs-cTnI,High-Sensitivity Cardiac Troponin I;CK-MB, Creatine Kinase Myocardial Band; NT-proBNP,N-terminal pro-brain natriuretic peptide;

**Table 3. Univariate and multivariate Logistic regression analysis of SFTS patients with concurrent HLH.**

| Parameters | Univariate | | | Multivariate | | |
|---|---|---|---|---|---|---|
| | OR | 95%CI | P | OR | 95%CI | P |
| Diarrhea | 0.496 | [0.254,0.966] | 0.039 | 0.434 | [0.211,0.895] | **0.024** |
| Platelet | 0.969 | [0.953,0.985] | <0.001 | 0.966 | [0.947,0.985] | **0.001** |
| ALT/AST ratio | 0.179 | [0.038,0.835] | 0.029 | 0.016 | [0.001,0.338] | **0.008** |
| APTT | 1.014 | [1.004,1.025] | 0.008 | 1.019 | [0.997,1.041] | 0.097 |
| Fibrinogen | 0.587 | [0.368,0.934] | 0.025 | 0.630 | [0.366,1.087] | 0.097 |
| DD | 1.101 | [1.054,1.151] | <0.001 | 1.088 | [1.033,1.146] | **0.001** |
| Time from onset to admission | 0.894 | [0.805,0.992] | 0.035 | 0.907 | [0.813,1.012] | 0.081 |
| LDH | 1.001 | [1.000,1.001] | <0.001 | 1.001 | [1.001,1.002] | **0.001** |
| Ferritin | 1.000 | [1.000,1.000] | <0.001 | 1.000 | [1.000,1.000] | **0.002** |
| IL-10 | 1.002 | [1.000,1.004] | 0.030 | 1.001 | [0.999,1.004] | 0.363 |

patients was also higher in those without HLH, with a statistically significant difference in mortality (HR = 1.635, 95% CI [0.967-2.764], *P* = 0.031), as shown in Fig 1. Among the 53 SFTS patients with concurrent HLH, they were categorized into a survival group (28 cases) and a death group (25 cases) based on outcomes. The demographic characteristics, clinical presentations at admission, and laboratory parameters of the two groups are detailed in S1 Table. Univariate and multivariate logistic regression analyses of differential indicators at admission suggested that age, FIB, and PCT are independent risk factors for mortality in SFTS patients with concurrent HLH, with specific results shown in Table 4.

Further analysis using ROC curves revealed significant findings. Age (AUC = 0.804), FIB (AUC = 0.741), and PCT (AUC = 0.761) were important indicators for mortality in SFTS patients with concurrent HLH. The combination of these three indicators demonstrated good diagnostic value for predicting mortality in SFTS patients with concurrent HLH (AUC = 0.903,Fig 1B). Based on the ROC curve, the optimal cutoff values for each indicator were calculated based on the survival outcomes of the patients(as shown in S2 Table), and the data was grouped. Kaplan-Meier survival curve

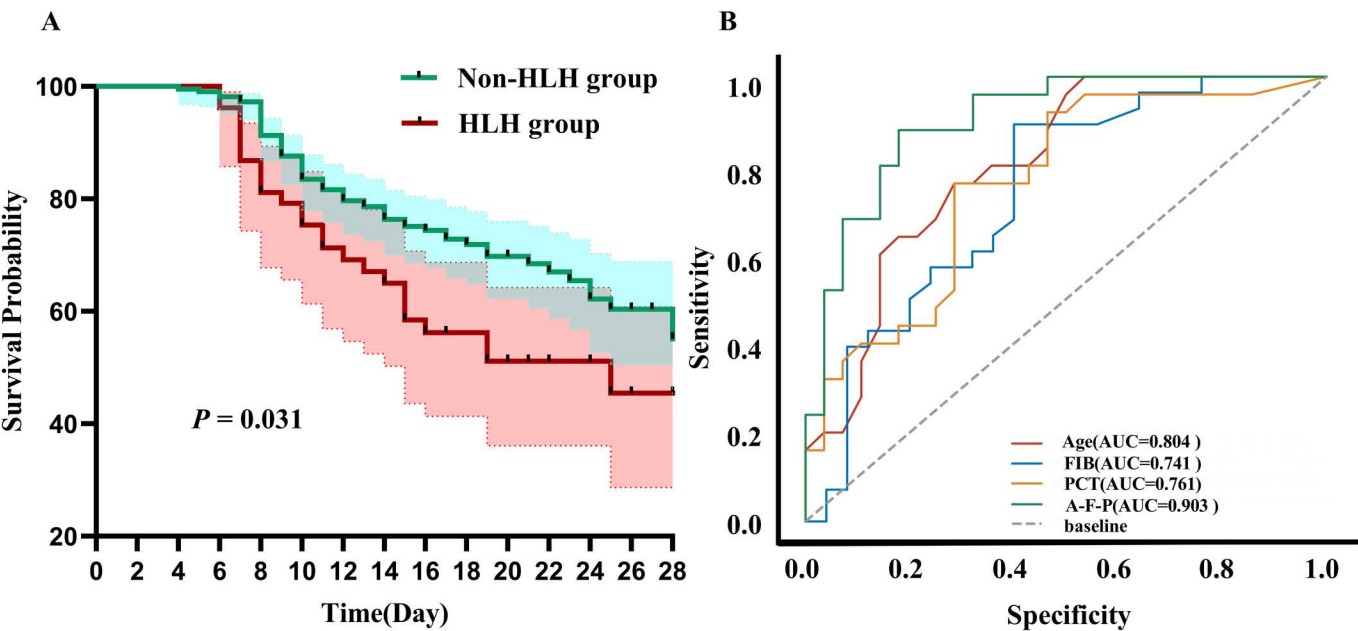

**Fig 1. (A) Kaplan-Meier survival curves showing 28-day survival rates between the two groups;(B) ROC analysis of death risk factors in SFTS patients with HLH.** Abbreviations: AUC, area under the curve; A-F-P, combination of Age, Fib (fibrinogen), and PCT (procalcitonin).

**Table 4. Univariate and multivariate Logistic regression analysis of prognosis of SFTS patients with HLH.**

| Parameters | Univariate | | | Multivariate | | |
|---|---|---|---|---|---|---|
| | OR | 95%CI | *P* | OR | 95%CI | *P* |
| Age | 1.159 | [1.064,1.261] | 0.001 | 1.208 | [1.073,1.359] | 0.002 |
| FIB | 0.278 | [0.096,0.808] | 0.019 | 0.155 | [0.038,0.636] | 0.010 |
| PCT | 1.804 | [1.029,3.163] | 0.039 | 2.990 | [1.041,8.588] | 0.042 |
| ALT/AST ratio | 0.016 | [0.0,0.602] | 0.025 | 0.020 | [0.000,2.578] | 0.115 |

analysis of the 28-day survival rate indicated that age ≥ 64 years (*P* = 0.001, Fig 2A), FIB ≤ 2.23 g/L (*P* = 0.023, Fig 2B), and PCT ≥ 0.9 ng/ml (*P* = 0.009, Fig 2C) are associated with poor prognosis.

## Dynamic characteristics of risk factors for mortality in SFTS patients with HLH

We further observed the dynamic changes in risk factors from admission to 20 days of hospitalization in SFTS patients with concurrent HLH. In the death group, PCT levels continued to rise after admission, whereas in the survival group, PCT levels gradually increased after admission but started to decline around day 8 post-admission, as shown in Fig 3A. The FIB levels in the death group consistently decreased after admission and fell below the reference range, while the survival group's FIB levels slightly decreased after admission and then gradually rose, remaining within the normal reference range, as shown in Fig 3B. We also observed the ferritin levels in SFTS patients with concurrent HLH. Apart from differences in the levels, both groups showed an initial increase followed by a decrease. The results are shown in S1 Fig.

## Discussion

SFTS is a severe viral disease with high morbidity and mortality. HLH, a rare and life-threatening condition, can be triggered by various viruses, including human herpesvirus, hemorrhagic fever viruses (dengue, Ebola, hantavirus), human

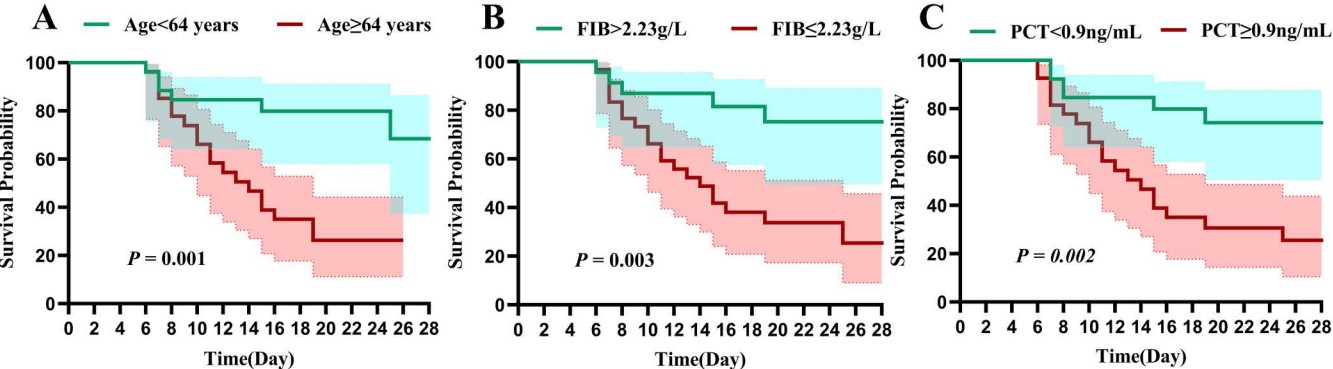

**Fig 2. Kaplan-Meier survival curves demonstrated 28-day survival rates in SFTS patients with HLH based on varying levels of (A) age, (B) FIB, and (C) PCT.**

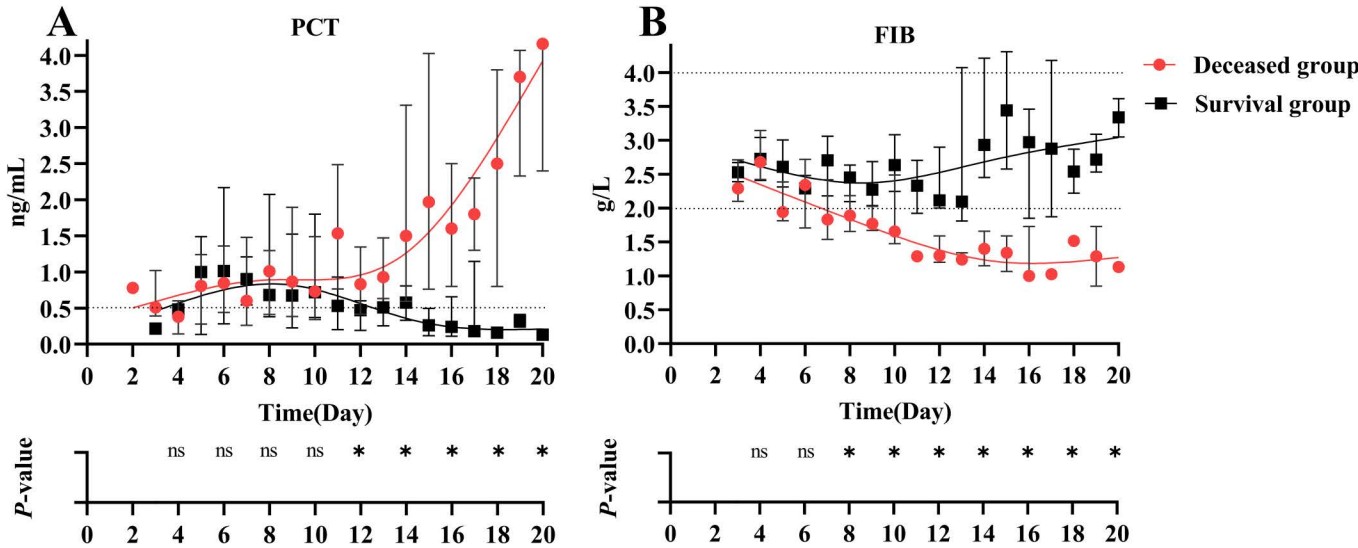

**Fig 3. Dynamic characteristics of (A) FIB and (B) PCT between the survival and deceased groups of SFTS patients with HLH.** The dotted line represents the biological reference interval.The biological reference range for PCT is less than 0.5 ng/mL; the biological reference range for FIB is 2-4 g/L.Data are presented as median (IQR). ns = no significance, *P < 0.05.

immunodeficiency virus, and SARS-CoV-2 [13–14]. Reports indicate that SFTSV infection can also trigger HLH, leading to higher mortality [10]. Therefore, evaluating prognostic factors available at patient admission is crucial for clinicians to make timely and effective management decisions.

In this study, we conducted a comprehensive analysis of the clinical features and laboratory markers of SFTS patients. Our findings suggest that diarrhea may reduce the risk of HLH in SFTS patients (OR=0.434). Previous research has reported a higher prevalence of diarrhea among surviving SFTS patients [15], whereas other studies have shown that diarrhea increases the risk of death in SFTS patients [16]. Therefore, the role of diarrhea in SFTS remains controversial. Further examination of the prognosis of SFTS patients with HLH showed no significant difference in the presence of diarrhea symptoms at admission between survivors and non-survivors. We propose that this finding may be due to the shorter duration from symptom onset to hospitalization in cases complicated by HLH, along with prompt fluid infusion therapy

following admission, which may have alleviated prognostic differences. In the analysis of laboratory markers, SFTS patients with HLH exhibited significant differences in blood routine markers, liver function markers, coagulation markers, inflammatory indicators, and cytokine markers. HLH patients typically exhibit pancytopenia due to cytokine storm, macrophage activation, bone marrow suppression, and increased hemophagocytosis [17–19], which explains why SFTS patients complicated by HLH demonstrate significantly lower leukocyte, lymphocyte, monocyte, and platelet counts compared to non-HLH cases. Critical SFTS patients can rapidly develop into multiple organ dysfunction syndrome (MODS) and disseminated intravascular coagulation (DIC) [5]. This study also suggests that ALT, AST, LDH, and DD are significantly increased in SFTS patients with HLH. In SFTS patients, a decreased ALT/AST ratio and elevated LDH levels typically reflect the extent of tissue injury and cellular destruction, which are considered as markers of poor prognosis in SFTS patients [20–21]. Notably, SFTS patients with HLH exhibit higher ALT, AST, and LDH levels, along with a lower ALT/AST ratio, all of which collectively signal worse prognosis. In patients with HLH upon admission, the elevation of APTT, TT, and DD, as well as the decrease of FIB, may be related to the inflammatory response and excessive activation of the immune system in SFTS patients [22]. In SFTS patients complicated with HLH, the coagulation abnormalities may present dynamic evolutionary characteristics: in the early stage, the coagulation indexes are mainly characterized by hypocoagulation; however, with the aggravation of endothelial injury, microvascular thrombosis may occur in the later stage, ultimately progressing to DIC [23]. This study shows that a decreased ALT/AST ratio, elevated LDH, and increased DD are independent risk factors for HLH in SFTS patients. Ferritin and PCT, as inflammatory biomarkers, demonstrate significant clinical value in the early prognostic assessment of SFTS patients [24]. Studies have confirmed that elevated PCT levels are significantly associated with increased mortality in SFTS patients, particularly during the acute phase of the disease, where changes in PCT levels can serve as important indicators for evaluating prognosis [20,24]. Notably, extremely elevated ferritin levels in HLH patients have been clearly identified as predictors of disease severity and poor prognosis [25]. The results of this study indicate that SFTS patients complicated with HLH have significantly higher ferritin and PCT levels compared to non-HLH patients, suggesting that ferritin and PCT may hold important clinical significance in disease severity and prognosis. However, the multivariate regression analysis suggested no correlation between ferritin levels and the occurrence of HLH in SFTS patients. Some studies argue that the diagnostic value of serum ferritin levels for HLH in adults may not be clear, as other conditions can also activate macrophages to produce ferritin [26]. A study establishing an SFTSV infection model in mice also indicated significant increases in megakaryocytes in the spleen and bone marrow early in the infection [27]. Additionally, an autopsy of a deceased SFTS patient showed significant infiltration of activated macrophages and increased hemophagocytes in the liver, spleen, and bone marrow, suggesting that ferritin levels could significantly increase in the early stage of of SFTS infection [11]. Research indicates that high serum IL-6 and IL-10 levels at admission are independent risk factors for in-hospital mortality in SFTS patients [28]. Notably, when SFTS is accompanied by HLH, the combined rise in IL-6, IL-2R, and IL-10 can worsen the cytokine storm [29]. Thus, monitoring cytokines during the acute phase may help guide immunomodulatory therapy to control inflammation and evaluate patient prognosis..

Some case reports suggest that concurrent HLH is a critical factor for mortality in SFTS patients [3,18]. This study further analyzes the prognosis of SFTS patients with concurrent HLH, confirming an increased mortality rate in these patients. Previous studies have established that age is an independent risk factor for mortality in SFTS patients, potentially related to lower immune function, higher infection rates, and higher complication rates in the elderly [16,30]. Additionally, reports indicate that age is associated with the prognosis of HLH patients [31], consistent with our findings. Studies have shown that the PCT level at admission is a key indicator for predicting the prognosis of SFTS patients, and its increase is usually related to bacterial infection, one of the main causes of death in SFTS patients [11,32]. In SFTS patients, the elevated PCT level may be caused by multiple factors, including immune system overactivation, systemic inflammatory response following HLH, and bacterial infection itself. Beyond systemic inflammation, PCT may worsen SFTS by activating coagulation pathways, with research linking elevated PCT levels to coagulation dysfunction in these patients [33]. In COVID-19 cases, higher PCT levels correlate with disease severity and coagulopathy, potentially

indicating DIC [34–35]. SFTS-associated DIC patients also show increased inflammatory markers such as IL-6 and CRP, which, like PCT, may be crucial in DIC development [36]. This study suggests that PCT may be an independent risk factor for death in SFTS patients with HLH. Moreover, fibrinogen, as an independent risk factor for the prognosis of HLH patients, may drive fulminant hyperinflammatory response syndrome after HLH with SFTS patients, resulting in liver dysfunction, DIC, and fibrinolysis [37]. Therefore, monitoring the levels of PCT and fibrinogen is of great significance for evaluating the prognosis of SFTS patients and guiding clinical treatment.

To confirm the relationship between FIB and PCT levels and the prognosis of SFTS patients with HLH. We found that the mortality rate was higher in patients older than 64 years, with FIB levels ≤ 2.23 g/L and PCT levels ≥ 0.9 ng/ml at admission. The combination of these three indicators could predict the prognosis of patients effectively. Further observation of the dynamic changes in FIB and PCT revealed significant differences between the death and survival groups in SFTS patients with HLH.In the survival group, FIB decreased slightly after admission and then increased within the reference range. However, in the death group, FIB levels continued to decline, consistent with existing research reports [38]. Additionally, the PCT level in the death group continued to rise, which has also been observed in other studies when observing deceased SFTS patients [24]. Therefore, continuous monitoring of the two indicators during hospitalization could provide valuable insights into disease progression and patient prognosis.

This study also has some limitations.Although the HLH-2004 criteria remain the most widely used standard for HLH diagnosis, they have certain limitations. Parameters such as hemophagocytosis, serum NK cell activity, and sCD25 concentration are not routinely measured, which may lead to underdiagnosis. Additionally, variability in physicians' diagnostic criteria and documentation practices may introduce potential bias. Some SFTS patients, particularly those with complex presentations or in early stages of the disease, might not meet all diagnostic criteria, resulting in missed diagnoses; The retrospective nature of this study may not capture all relevant variables. For instance, the duration and severity of diarrhea during hospitalization were not consistently recorded, which could influence outcome assessments; The unequal sample sizes between SFTS patients with and without HLH might affect the statistical power of our comparisons. Although we adjusted for known confounders using multivariate regression, residual confounding may persist. Future studies with larger cohorts are needed to address this limitation; Retrospective data may not fully account for dynamic variables such as viral load fluctuations [39,40] and individual immune status [41,42], which could impact HLH development and prognosis in SFTS patients. Prospective studies are warranted to better control for these factors; Geographic differences in viral strains and genotypes may influence SFTS outcomes [43]. Whether specific genotypes affect HLH incidence or prognosis in SFTS patients remains unclear and requires further investigation.

## Conclusions

We found that a decreased ALT/AST ratio, increased LDH levels, and elevated DD levels are independent risk factors for HLH in SFTS patients. Age, FIB, and PCT are independent risk factors for mortality in SFTS patients with HLH. Combining these three indicators can effectively predict patient prognosis. Patients aged ≥64 years, with FIB ≤ 2.23 g/L and PCT ≥ 0.9 ng/mL at admission, have a higher mortality rate. Additionally, dynamic monitoring of FIB and PCT levels can better reflect disease progression, aiding doctors in timely intervention and reducing patient mortality.

## Supporting information

**S1 Table. Comparison of clinical characteristics and laboratory markers at admission for SFTS patients with HLH between the Surviving and Deceased Groups.**
(DOCX)

**S2 Table. ROC Curve Analysis for Predicting Death in SFTS Patients with HLH.**
(DOCX)

**S1 Fig. Fig Dynamic characteristics of ferritin between the survival and deceased groups of SFTS patients with HLH.**

(TIF)

## Acknowledgments

The authors thank all the donors and patients for participating in this study.

## Author contributions

**Data curation:** Bo Zhang.

**Formal analysis:** Jia-le Gong, Qin Liao.

**Investigation:** Jia-le Gong.

**Methodology:** Bo Zhang, Qin Liao.

**Visualization:** Bo Zhang.

**Writing – original draft:** Bo Zhang.

**Writing – review & editing:** Hao-long Zeng.

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
