## [Decision Letter · Decision Letter 0]

Jun 01 2025

PNTD-D-24-01931

Unveiling Fatal Complications: Predicting Hemophagocytic Lymphohistiocytosis in SFTS Patients

Dear Dr. Zhang,

Thank you for submitting your manuscript to PLOS Neglected Tropical Diseases. After careful consideration, we feel that it has merit but does not fully meet PLOS Neglected Tropical Diseases's publication criteria as it currently stands. Therefore, we invite you to submit a revised version of the manuscript that addresses the points raised during the review process.

Please submit your revised manuscript within 60 days Jun 01 2025 11:59PM. If you will need more time than this to complete your revisions, please reply to this message or contact the journal office at plosntds@plos.org. Please include the following items when submitting your revised manuscript:

We look forward to receiving your revised manuscript.

Kind regards,

Richard A. Bowen, DVM PhD

Academic Editor

Andrea Marzi

Section Editor

Shaden Kamhawi

co-Editor-in-Chief

Paul Brindley

co-Editor-in-Chief

**Additional Editor Comments:**

Thank you for your submission. Your manuscript has been reviewed by three experts and they offer suggestions for improvement. Please evaluate their comments, edit your manuscript and respond to those comments. We look forward to seeing a revised version.

**Journal Requirements:**

At this stage, the following Authors/Authors require contributions: Bo Zhang, Jia-le Gong, Hao-long Zeng, and Qin Liao. Please ensure that the full contributions of each author are acknowledged in the "Add/Edit/Remove Authors" section of our submission form.

4) In the online submission form, you indicated that "The data for this study can be made available upon reasonable request to the corresponding author, Due to privacy and confidentiality concerns, the relevant datasets are not publicly available.". All PLOS journals now require all data underlying the findings described in their manuscript to be freely available to other researchers, either

- In a public repository

- Within the manuscript itself

- Uploaded as supplementary information.

**Reviewers' Comments:**

Reviewer's Responses to Questions

**Key Review Criteria Required for Acceptance?**

**Methods:**

-Are the objectives of the study clearly articulated with a clear testable hypothesis stated?

-Is the study design appropriate to address the stated objectives?

-Is the population clearly described and appropriate for the hypothesis being tested?

-Is the sample size sufficient to ensure adequate power to address the hypothesis being tested?

-Were correct statistical analysis used to support conclusions?

-Are there concerns about ethical or regulatory requirements being met?

Reviewer #1: (No Response)

Reviewer #2: (No Response)

Reviewer #3: The objectives for this paper – identifying predictive biomarkers for HLH and markers for. HLH mortality – are clearly stated. There are no citations provided for the methods section, however, and therefore it is difficult to ascertain whether the methods used were appropriate or if they were novel to the manuscript. Additionally, in 102, the author refers to a “big data platform”, please provide the specific data platform. The n-size for the total population and the population that were HLH positive appear appropriate to address the objective of this manuscript.

**Results:**

-Does the analysis presented match the analysis plan?

-Are the results clearly and completely presented?

-Are the figures (Tables, Images) of sufficient quality for clarity?

Reviewer #1: (No Response)

Reviewer #2: (No Response)

Reviewer #3: The analysis presented matches the analysis plan. The results are clear and completely presented. The figures do appear slightly blurry, and this should be fixed before publication, but the content of the figures is clear.

**Conclusions:**

-Are the conclusions supported by the data presented?

-Are the limitations of analysis clearly described?

-Do the authors discuss how these data can be helpful to advance our understanding of the topic under study?

-Is public health relevance addressed?

Reviewer #1: (No Response)

Reviewer #2: (No Response)

Reviewer #3: The conclusions drawn are clearly supported by the data presented. The authors additionally contextualize their findings within the larger body of research or attempt to explain how their work disagrees with prior studies (i.e., the diarrhea results). The authors do discuss some limitations to their work, and while some are very clear, such as markers that could have been studied, some are far too vague. Specifically, the authors should expand on the phrase “influencing factors may not have been included”. Overall, this manuscript has significant public health relevance, as it provides critical diagnostic criteria and relationships between them to predict HLH.

**Editorial and Data Presentation Modifications?**

Reviewer #1: ※Minor comments:

1.What is the full name of APTT in line 113?

2.It is necessary to correct the usage of spacing in the manuscript, as there were many errors (line 114, 121, 123, 125, 127, 151~155, Table 1~3, 166, 169, 171, 190~195, and etc…). For example: in table 1 '(n=272)', the letter 'n' should be italicized, and there should be a single space between each character like ‘(n = 272)’.

Reviewer #2: (No Response)

Reviewer #3: In general, the paper should be reviewed for grammatical errors, such as in Line 90, which should say “We gathered hospitalized SFTS patients…”

**Summary and General Comments:**

Reviewer #1: ※Major comments:

1.The authors of this study provide insufficient explanation regarding the changes in the blood coagulation mechanism associated with increased procalcitonin (PCT) levels. The hypothesis that PCT elevation worsens the prognosis of SFTS patients by enhancing systemic inflammation lacks sufficient supporting evidence. Providing a discussion on the correlation between PCT and disseminated intravascular coagulation (DIC) would strengthen the argument. Additional explanation in the Discussion section is essential.

2."In Table 1&2, which compares non-HLH and HLH patients, the HLH-classified group shows increased APTT and TT, decreased fibrinogen, and increased D-dimer, all indicating reduced blood coagulation activity. This could suggest that SFTS patients diagnosed with HLH may have impaired coagulation function, potentially leading to a lower incidence of DIC. However, as the authors mentioned in lines 268–269, critical SFTS patients are at high risk due to progression to DIC and MODS, which seems contradictory. Furthermore, as seen in the outcome data, patients classified as HLH show a significantly higher mortality rate. The authors appear to have skipped an in-depth discussion of this discrepancy and moved directly to analyzing risk factors in HLH-classified SFTS patients. A more thorough discussion of Table 1&2 is essential to provide a clearer interpretation of these findings.

3.Based on the title of the paper, one might anticipate that the research focuses on distinguishing HLH among SFTS patients. However, the precise final conclusion of this study is the identification of important factors for assessing the risk of SFTS patients classified as HLH. Therefore, I believe it is necessary to revise the title.

Reviewer #2: (No Response)

Reviewer #3: Zhang and their collaborators set out to identify biomarkers to predict HLH in SFTS patients and, further, predict mortality amongst HLH patients. This work involved comprehensive blood panels and statistical analysis on 272 SFTS patients from Tongji Hospital. Their analysis identified that platelet counts, ALT/AST ratios, LDH, and DD were all predictive risk factors for HLH. They also determined that Age, fibrinogen (FIB), and procalcitonin (PCT) were all strong predictors of mortality due to HLH. These results allowed the authors to establish diagnostic thresholds that can be used by other doctors when treating SFTS and SFTS-HLH patients. On its face, this paper is a significant step forward towards understanding and combating SFTS. However, this paper is not without issues, some of which are major and some minor, that must be addressed before it is accepted. The primary issue that requires significant revisions is that the authors failed to provide any citations within the methods section. I am unable to determine whether the methods used are novel or adapted from existing work due to the absence of citations, which makes it impossible to evaluate the study's originality and reproducibility. The secondary issue is that the limitation section, particularly when discussing confounding factors, is too vague to be interpreted appropriately. Once these two issues are addressed, I see no reason why this manuscript should not be accepted.

PLOS authors have the option to publish the peer review history of their article (what does this mean? ). If published, this will include your full peer review and any attached files.

**Do you want your identity to be public for this peer review?** For information about this choice, including consent withdrawal, please see our Privacy Policy .

Reviewer #1: No

Reviewer #2: No

Reviewer #3: No

**Figure resubmission:**
---

## [Editor Report · Decision Letter 1]

Jul 03 2025

PNTD-D-24-01931R1Unveiling Fatal Risk Factors: Predicting Hemophagocytic Lymphohistiocytosis in SFTS PatientsPLOS Neglected Tropical Diseases Dear Dr. Zhang, Thank you for submitting your manuscript to PLOS Neglected Tropical Diseases. After careful consideration, we feel that it has merit but does not fully meet PLOS Neglected Tropical Diseases's publication criteria as it currently stands. Therefore, we invite you to submit a revised version of the manuscript that addresses the points raised during the review process.

Please submit your revised manuscript within 30 days Jul 03 2025 11:59PM. If you will need more time than this to complete your revisions, please reply to this message or contact the journal office at plosntds@plos.org. Please include the following items when submitting your revised manuscript:

* A rebuttal letter that responds to each point raised by the editor and reviewer(s). You should upload this letter as a separate file labeled 'Response to Reviewers '. This file does not need to include responses to any formatting updates and technical items listed in the 'Journal Requirements' section below.

* A marked-up copy of your manuscript that highlights changes made to the original version. You should upload this as a separate file labeled 'Revised Manuscript with Track Changes '.

* An unmarked version of your revised paper without tracked changes. You should upload this as a separate file labeled 'Manuscript '.

We look forward to receiving your revised manuscript.

Kind regards,

Richard A. Bowen, DVM PhD

Academic Editor

Andrea MarziSection EditorPLOS Neglected Tropical Diseases

Shaden Kamhawi

co-Editor-in-Chief

Paul Brindley

co-Editor-in-Chief

**Additional Editor Comments :** Thank you for modifying your manuscript to address reviewer comments - I think we both agree that it is significantly improved. I would ask that you make two further minor revisions if possible:

1) Figures 1 and 2 remain low quality and will not be good for publication - are you able to re-create them in a higher quality form? The new Figure 3 looks much improved.

2). The precision of numbers in tables is excessive and unnnecessary. For example: Table 1, first 2 lines have Females as 148 (54.412) and Males as 45.558. Please change these to 54.4 and 45.6% respectively. Please do the same for all of the numbers presented in your tables, restricting all figures to 1 number after the decimal point. Thank you - that will improve readability. **Journal Requirements:**

1) Thank you for stating "All authors declare that there is no conflict interest." If you have no competing interests to declare, please state "The authors have declared that no competing interests exist"

---

## [Editor Report · Decision Letter 2]

Dear Mr Zhang,

We are pleased to inform you that your manuscript 'Unveiling Fatal Risk Factors: Predicting Hemophagocytic Lymphohistiocytosis in SFTS Patients' has been provisionally accepted for publication in PLOS Neglected Tropical Diseases.

Best regards,

Richard A. Bowen, DVM PhD

Academic Editor

Andrea Marzi

Section Editor

Shaden Kamhawi

co-Editor-in-Chief

Paul Brindley

co-Editor-in-Chief

Thank you for modifying your manuscript in response to reviewer and editor suggestions. It will be a valuable contribution to the field.